# Lignin-Only Polymeric Materials Based on Unmethylated Unfractionated Kraft and Ball-Milled Lignins Surpass Polyethylene and Polystyrene in Tensile Strength

**DOI:** 10.3390/molecules24244611

**Published:** 2019-12-17

**Authors:** Yi-ru Chen, Simo Sarkanen, Yun-Yan Wang

**Affiliations:** 1Department of Bioproducts and Biosystems Engineering, University of Minnesota, St. Paul, MN 55108, USA; chenx137@umn.edu; 2Department of Forestry, Wildlife, and Fisheries, Center for Renewable Carbon, University of Tennessee, Knoxville, TN 37996, USA; ywang226@utk.edu

**Keywords:** lignin valorization, lignin blends, lignin configuration, lignin-based plastics.

## Abstract

Functional polymeric materials composed solely of lignin preparations appeared only very recently. A gradual paradigm shift spanning 56 years has revealed how lignin–lignin blends can upgrade the performance of 100 wt% lignin-based plastics. The view, first espoused in 1960, that lignin macromolecules are crosslinked reduces the plausibility of creating functional polymeric materials that are composed only of lignin preparations. Lignin-based materials would be much weaker mechanically if interstices remain in significant numbers between adjoining macromolecular structures that consist of rigid crosslinked chains. In 1982, random-coil features in the hydrodynamic character of kraft lignin (KL) components were evident from ultracentrifugal sedimentation equilibrium studies of their SEC behavior. In 1997, it was recognized that the macromolecular species in plastics with 85 wt% levels of KL are associated complexes rather than individual components. Finally, in 2016, the first polymeric material composed entirely of ball-milled softwood lignin (BML) was found to support a tensile strength above polyethylene. Except in its molecular weight, the BML was similar in structure to the native biopolymer. It was composed of associated lignin complexes, each with aromatic rings arranged in two domains. The inner domain maintains structural integrity largely through noncovalent interactions between cofacially-offset aromatic rings; the peripheral domain contains a higher proportion of edge-on aromatic-ring arrangements. Interdigitation between peripheral domains in adjoining complexes creates material continuity during casting. By interacting at low concentrations with the peripheral domains, non-lignin blend components can improve the tensile strengths of BML-based plastics to values well beyond those seen in polystyrene. The KL-based plastics are weaker because the peripheral domains of adjoining complexes are less capable of interdigitation than those of BML. Blending with 5 wt% 1,8-dinitroanthraquinone results in a tensile strength above that of polyethylene. Analogous effects can be achieved with 10 wt% maple γ-valerolactone (GVL) lignin which, with a structure close to the native biopolymer, imparts some native character to the peripheral domains of the KL complexes. Comparable enhancements in the behavior of BML complexes upon blending with 10 wt% ball-milled corn-stover lignin (BMCSL) result in lignin-only polymeric materials with tensile strengths well beyond polystyrene.

## 1. Introduction

Adding sufficient value to coproduct lignins remains the most prominent obstacle to securing profitability in biorefineries that intend to produce liquid fuels and commodity organic chemicals from lignocellulosic plant materials. In the excellent 17-chapter compendium describing emerging approaches to lignin valorization that appeared in 2018 [1], lignin-based plastics were conspicuously absent. The most productive approach to creating functional polymeric materials with very high lignin contents is largely determined by the configuration(s) of the lignin derivatives from which they will be made. However, all-too-common descriptions of lignin such as the 2018 picture of “a complex and irregular biopolymer of crosslinked phenylpropanoid units” [2] imply that success may only be achieved by rearranging or in some way counterbalancing lignin’s skeletal structure. 

In 2016, a softwood ball-milled lignin, on its own, was found to be capable of forming a plastic with tensile strength above that of polyethylene [3]. In the present work, softwood kraft lignin (Indulin), when blended with just 5 wt% 1,8-dinitroanthraquinone, is shown to form another lignin-based polymeric material with tensile strength greater than polyethylene. It is natural to ask why such results were not reported 40 years ago.

### 1.1. Macromolecular Lignin Configuration: Original Impressions

A problem arose in a disagreement regarding two opposing ideas about macromolecular lignin configuration that were first aired between 1955 and 1960. Owing to their contrasting implications, it is instructive to consider a brief summary of the salient experimental work.

In a 1955 paper, the molecular weights (*M*) of seven 12–120 kDa softwood ligninsulfonate fractions in aqueous 1.0 M NaCl were measured by Rayleigh light scattering using a protocol cognizant of the need to take solute fluorescence and absorbance into account [4]. The corresponding diffusion coefficients (*D*) were determined in an aqueous 0.02 M NaCl solution-to-gel apparatus; the relationship with respect to molecular weight ($D\propto M^{-0.57}$) was consistent with a non-free-draining random coil description of ligninsulfonate chains [4].

Nine years later, another study of four 43–126 kDa softwood ligninsulfonate fractions in aqueous 0.10 M NaOH provided confirmatory evidence in a value of −0.49 for the molecular-weight exponent describing the dependence of the diffusion coefficient (*D*) on *M* [5]. In these comparative ultracentrifuge and light-scattering studies, the reliability of the molecular-weight data was confined to four fractions that did not form a significant gel layer in the ultracentrifuge cell during approach to sedimentation equilibrium [5].

Meanwhile, in 1957 it was revealed that, when immersed in water or aqueous alkali, softwood periodate lignin can absorb components from solution and swell without undergoing dissolution. This behavior was taken to favor a three-dimensional network model as a more reasonable representation of macromolecular lignin configuration [6], although experimental support for this contention was unclear. 

Three years later in 1960, it had become evident that 50–48,000 kDa fractions of alkali lignin produced (by 15-min treatment with aqueous 2.5 M NaOH at 162 °C) from softwood periodate lignin did not exhibit behavior consistent with a three-dimensional network configuration for lignin macromolecules [7]. At pH 9.65 the changes in intrinsic viscosity ([*η*]) and ultracentrifugal sedimentation-velocity coefficient (*s_w_*) with weight-average molecular weight (light-scattering *M_w_*) both yielded relationships ($\left[\eta \right]\propto \;M_{w}^{0.32}\;\mathrm{and}\;s_{w}\;\propto \;M_{w}^{0.52}$) that fell between those for a non-free-draining random coil and an Einstein sphere [7].

A contemporaneous study of ligninsulfonate fractions (prepared by sulfonating spruce periodate lignin) found that both the radius of gyration (*R_g_* from Rayleigh light scattering) and intrinsic viscosity ([*η*]) of the polyanionic components increase with decreasing ionic strength in a protocol involving iso-ionic dilution [8]. The relationship between the two parameters ($\left[\eta \right]\propto \;R_{g}{}^{3}$) could not distinguish between the swelling behavior of a macromolecular crosslinked microgel and a non-free-draining random coil [8]. It was pointed out that the onset of free-draining character with random-coil expansion could shift the relationship toward $\left[\eta \right]\propto R_{g}{}^{2}$, but there was no evidence that this was actually occurring. 

Despite the overall equivocality of the experimental evidence, a crosslinked macromolecular chain was staunchly favored as the more likely configuration for a ligninsulfonate component, the skeletal structure of which was depicted as a microgel (Figure 1). This hypothesis has persisted in many quarters until the present day [2]. It has reinforced a vision of rigidity in lignin chains that has profoundly influenced decisions about how functional lignin-based polymeric materials can best be formulated.

In particular, it was assumed that the only way to avoid the brittleness of polymeric materials with high levels of rigid lignin domains would be to introduce (by covalent bonding or noncovalent blending) an adequate proportion of soft segments. This approach usually led to a 40 wt% incorporation limit for lignin in hopefully functional polymeric materials [9]. Such trends are encountered when lignin derivatives undergo covalent crosslinking (Figure 2) [10,11] and when they contribute to multiphase blends [12]. 

### 1.2. Macromolecular Lignin Configuration: Current Perspectives

From studies capable of reliable discrimination, no evidence has emerged to suggest that macromolecular lignin components embody (long-chain) branching or crosslinking. For example, the verdict from size-exclusion chromatographic (SEC) absolute-molecular-weight calibration curves is clear. Beyond the excluded limit of a Sephadex G100 column, plots of log *M_w_* versus aqueous 0.10 M NaOH elution volume (Figure 3) for paucidisperse hardwood and softwood kraft lignin fractions [13,14] are parallel to those of poly(styrenesulfonate). Moreover, the polydispersity (reflected by *M_z_*/*M_w_*) of paucidisperse softwood kraft lignin fractions in aqueous 0.10 M NaOH does not increase with their average molecular weight [15]. Thus, kraft lignin components behave hydrodynamically as though they possess appreciably expanded random coil conformations. It is therefore very unlikely that long-chain branching or crosslinking is present in native hardwood or softwood lignin macromolecules.

### 1.3. Plastics with 100 wt% Lignin-Derivative Contents

Polymeric materials composed solely of (native) softwood ball-milled lignin (BML) can exhibit tensile strengths (Figure 4) that surpass those of polyethylene (30 MPa [16]). Moreover, the methylation of BML results in a material (MBML) that, on its own, approaches polystyrene (46 MPa [16]) in tensile strength (Figure 4). To maintain integrity, these lignin-based plastics rely on strong noncovalent intermolecular interactions that range from ~4 kcal/mol between edge-on aromatic rings to 7–11 kcal/mol for cofacially-offset aromatic-ring arrangements [17]. In the absence of hydrogen bonding, lateral displacement between cofacially-offset aromatic rings occurs more readily, so MBML-based materials are usually less brittle than their BML-based counterparts. 

The X-ray powder diffraction patterns of unblended BML- and MBML-based materials consist of two overlapping Lorentzian peaks [3] reflecting (primarily) the distributions of separation distances between cofacially-offset and edge-on aromatic rings in the lignin components (Figure 5). The respective peak maxima occur at 4.0–4.2 Å and 5.6–6.1 Å equivalent Bragg spacings. 

The predominant macromolecular species in plastics with high lignin contents are well-defined associated complexes rather than individual molecular components. The cohesion of the more stable inner domains largely depends on cofacially-offset arrangements of the aromatic rings, while the peripheral domains accommodate a greater frequency of edge-on orientations [17]. It is through their peripheral domains that continuity between adjoining associated complexes is established to create mechanical strength during casting [17]. The accompanying alterations in the distributions of aromatic-ring arrangements are evident in changing contributions from the two Lorentzian peaks (Figure 5). Among the two categories, the new proportions of interacting aromatic rings in the peripheral domains of adjoining lignin complexes depend on how readily new cofacially-offset arrangements can be created between “edge-on” and “on-edge” aromatic rings at the surfaces where interdigitation between macromolecular entities occurs (Figure 6). Consequently, the distinguishable character of the peripheral domains may expand or contract in the space occupied within the body of the lignin-based plastic that is formed through casting (Figure 5).

Understanding how material continuity is established so as to avoid interstices between associated lignin complexes in plastics with very high lignin contents is a prerequisite for developing functional >90 wt% lignin-based polymeric materials. While methylated lignin preparations often produce stronger (less-brittle) plastics, the cost of derivatization usually precludes manufacturing processes that include alkylation of the starting material. Nevertheless, industrial coproduct lignins present significant challenges in regard to the development of useful eco-friendly plastics. For example, in 2005 it was reported that plastics with 80 wt% levels of an ethylated and/or methylated higher-molecular-weight industrial kraft lignin fraction are capable of matching polystyrene in tensile strength [18]. Yet it is only now that material formulations containing 90 wt% unmethylated Indulin AT (Ingevity Corp., S.C.) have improved to the point where they can surpass the tensile strength of polyethylene. Such results have been achieved with both non-lignin and other-lignin blend components. Analogous formulations based on unmethylated ball-milled softwood lignin can now generate even more promising materials that go beyond polystyrene in tensile strength. These unprecedented, and perhaps unexpected, developments have been secured by learning how to manipulate the associated complexes of which lignin-based polymeric materials are composed.

## 2. Results and Discussion

Traditional kraft pulp mills can be regarded as first-generation biorefineries. At elevated temperatures (~170 °C for 2 h, for example), strongly alkaline aqueous solution containing bisulfide is capable of removing most of the lignin from a lignocellulosic feedstock (softwood chips, for example) in a process that produces cellulosic fibers (“pulp”), soluble carbohydrates, and their derivatives. The resulting kraft lignin has undergone extensive chemical changes that bring about profound alterations in structure and physicochemical behavior relative to the native biopolymer [19]. Consequently, kraft lignin is almost exclusively used as a fuel in the recovery boiler which houses the rate-limiting step of the mill operation. Any operational changes (from increasing pulp production to generating carbohydrate-derived liquid fuels or platform chemicals) requires valorization of the co-product kraft lignin. 

Indeed, softwood kraft lignin has been recognized as a renewable “raw material for the future” [20]. The problem is that it is structurally far removed from the native biopolymer [19]. Nevertheless, enough of its polymer backbone has been preserved for associative/dissociative processes between kraft lignin components to occur in a remarkably well-defined manner in solution. At pHs within a fairly narrow aqueous range above 11.5, >100 gL^−1^ kraft lignin undergoes reversible association; conversely, reversible dissociation takes place at <1 gL^−1^ kraft lignin concentrations in aqueous 0.10 M NaOH. The apparent molecular-weight distributions of kraft lignin preparations that have been treated in this way are depicted in Figure 7 (after desalting by elution through Sephadex LH20 in aqueous 35% dioxane). The Sephadex G100/aqueous 0.10 M NaOH elution profiles exhibit a common (isosbestic) intersection point, demonstrating that all relevant physicochemical processes are occurring at the same rate [21]. The result is remarkable in view of the >20-fold range in molecular weight of the interacting kraft lignin species (Figure 7). Confirmatory evidence is revealed in the average product of the molecular weights, 〈$m_{c}m_{l}$〉, of the kraft lignin complexes and components that participate in these associative/dissociative processes. A plot of weight-average (*M_w_*) versus the reciprocal of number-average (1/*M_n_*) molecular weight (Figure 8) generates a curve, the slope of which at any point is given by $-2\langle m_{c}m_{l}\rangle$ [22]. It is hereby evident that the value, 1.01 × 10^7^, of 〈$m_{c}m_{l}$〉 characterizing the relationship between the associatively/dissociatively homologous kraft lignin samples (Figure 7) remains constant [21] as the degree of association changes quite steeply (Figure 8). In the absence of restrictions, 〈$m_{c}m_{l}$〉 would tend toward $M_{n}^{2}$ when productive associative interactions are occurring randomly [22]. Under such circumstances, 〈$m_{c}m_{l}$〉 for the kraft lignin samples in Figure 7 undergoing interconversion in aqueous alkaline solution would vary 3.5-fold instead of remaining constant (Figure 8). Clearly, the associated complexes in these samples are well-defined entities that can only associate with individual kraft lignin components in a selective manner. Thus, the peripheral domains in associated softwood kraft lignin complexes would be expected to encounter difficulties in aggregating with one another. 

### 2.1. Plastics with Unmethylated Kraft Lignin Contents above 90 wt%

The impact of the powerful noncovalent forces between aromatic substructures is encountered in many lignin derivatives, such as Organosolv [22] and kraft [23] lignins. Pronounced consequences appear in the mechanical properties of polymeric materials with high lignin contents where the structures of associated lignin complexes are preserved. These effects were initially observed in the first 85 wt% kraft-lignin-containing plastics that had been formulated by blending with 12.6% poly(vinyl acetate), 1.6% diethyleneglycol dibenzoate, and 0.8% indene [24]. The degree of association between the individual molecular components in the kraft lignin preparations had been modulated by incubation at different concentrations in aqueous alkaline solution [24]. As shown in Figure 9, the tensile behavior of the 85 wt% kraft-lignin-containing plastics is influenced markedly by the extent to which the kraft lignin complexes and components have undergone reversible association with one another [24]. Interestingly, the glass transition temperature (T_g_) of the materials (29.9–30.0 °C) changed very little with the degree of association of the kraft lignin preparations [24]. Consequently, these first unmethylated kraft-lignin-based plastics are best described as heterogeneous blends in which the macromolecular kraft lignin species play a dominant role in influencing mechanical behavior [24]. The tensile strength of these plastics is linearly dependent on kraft-lignin *M_w_* as it varies with the degree of association of the preparation. The important point is that the individuality of the associated complexes was not lost as material continuity in the kraft-lignin-rich domains developed.

While plastics composed solely of (native) ball-milled softwood lignin (BML) exhibit tensile strengths that surpass polyethylene (Figure 4), industrial softwood kraft lignin is much more challenging as an alternative starting material. On its own, it is weaker than BML, and so far non-lignin blend components in formulations that give rise to functional plastics with >90 wt% kraft-lignin contents have been non-traditional in nature (Figure 10). Almost all of the relatively flexible β–*O*–4ʹ substructures in softwood lignin are cleaved during kraft pulping. Consequently, the rigid peripheral domains remaining in associated kraft lignin complexes are restricted in their ability to take full advantage of all potential noncovalent attractive interactions between these macromolecular entities when in close proximity to one another. The conclusion from the first 85 wt% kraft-lignin-containing plastics (Figure 9) is that the associated complexes do not necessarily lose their individuality when they contribute to material continuity in kraft-lignin-rich domains. Accordingly, only those non-lignin blend components that enhance aromatic-ring mobility in the peripheral domains of associated kraft-lignin complexes are likely to generate functional plastics with very high kraft-lignin contents.

Industrial softwood kraft lignin alone forms a material with tensile strength (9.5 MPa) about 3-fold lower than that of polyethylene. However, blending with just 5 wt% 1,8-dinitroanthraquinone results in a formulation with a tensile strength (34 MPa) that surpasses polyethylene (Figure 10). This simple result leads to the prospect of profitability in producing functional kraft-lignin-based plastics, as follows. Industrial kraft lignin ($0.83 of commercial lignin [25] yields 0.95 kg of purified material) is thoroughly washed with water ($0.55/kg [26]), air-dried, and blended with 5 wt% miscible blend component ($0.03 for 1,8-dinitroanthraquinone [27] per kg blend). The resulting mixture is compounded ($0.21/kg [28]) to create a material blend with an expected production cost below $1.69 per kg. Even though this estimate does not take process optimization or economies of scale into account, it is less than half of the 2017 U.S. polystyrene ($3.64 per kg) trading price [29].

The question arises as to whether plastics with >90 wt% softwood kraft lignin contents must always include (possibly eco-unfriendly) non-lignin blend components in viable formulations. Actually, lignin-only (biodegradable) blends are also capable of generating functional polymeric materials. A noteworthy example of an other-lignin blend component may be found in maple γ-valerolactone (GVL) lignin which, when cast on its own, forms a plastic with tensile strength (33 MPa) greater than that of polyethylene (Figure 11). Its tensile behavior is very similar to that of the softwood ball-milled lignin [3] shown in Figure 4. Presumably, this arises from the fact that GVL lignins are not far removed from the corresponding native lignins in structure [30] except that their molecular weights have been reduced because of inter-unit (predominantly) β–*O*–4 cleavage. 

A blend of softwood kraft lignin with just 10 wt% maple GVL lignin (Figure 11) results, upon casting, in a polymeric material with 35 MPa tensile strength (almost 4-fold higher than for kraft lignin alone). Its proximity to native lignin structure presumably enables maple GVL lignin to restore the poorly functional peripheral domains of associated kraft-lignin complexes that, in the absence of non-lignin blend components, can no longer participate fully in interdigitation with neighboring complexes owing to extensive β–*O*–4 cleavage during kraft pulping. The impact of 10 wt% maple GVL lignin is very similar to that of 5 wt% 1,8-dinitroanthraquinone. Thus, lignin-only blend formulations deserve careful consideration as potential means for creating functional polymeric materials composed of lignins alone.

### 2.2. Plastics with Unmethylated Ball-Milled Lignin Contents above 95 wt%

At 95–98 wt% levels in homogeneous materials containing non-lignin blend components, ball-milled softwood lignin (BML) can form plastics exhibiting tensile behavior that clearly surpasses the 46 MPa strength [16] of polystyrene (Figure 12). Particularly noteworthy are the blends containing 5 wt% tetrabromobisphenol A (a flame retardant) and 2 wt% poly(ethylene oxide-*b*-1,2-butadiene-*b*-ethylene oxide) which reach tensile strengths of 53.5 MPa with 7.5% elongation-at-break and 58.5 MPa with 9% elongation-at-break, respectively. The performance of these plastics with 95–98 wt% BML contents (Figure 12) extends well past the 90–95 wt% kraft-lignin-based plastics that merely go beyond polyethylene (Figure 11). The difference arises from the fact that, in contrast to kraft lignin, many of the relatively flexible β–*O*–4 linkages are preserved in BML. Consequently, interacting peripheral domains on adjoining associated-complex boundaries can readily participate in interdigitation to create material continuity during casting (Figure 6).

Under the circumstances, it would be instructive to determine whether other lignin preparations, possessing structural features not far removed from the native biopolymer, may act as beneficial blend components with BML. This question is explored in Figure 13, where the tensile properties of a 90 wt% BML blend with 10 wt% ball-milled corn-stover lignin (BMCSL) extend to 52 MPa strength with 9% elongation-at-break. This is the strongest lignin-only polymeric material hitherto reported: it easily exceeds the tensile strength of polystyrene. The result predicts that eco-friendly plastics with lignin contents approaching 100 wt% are likely to play an important part in achieving the profitable conversion of lignocellulose to commodity organic chemicals and functional polymeric materials.

## 3. Materials and Methods 

### 3.1. Materials

Ball-milled lignin (BML) was isolated from Jack pine as previously described [17]. Ball-milled corn-stover lignin (BMCSL) was provided by R. Katahira and D.K. Johnson, NREL. Acetylated BMCSL manifested a weight-average molecular weight (*M_w_*) of 5900 with polydispersity index 3.7. 

The *M_w_* of the softwood kraft lignin commercially available as Indulin (Ingevity Corp., North Charleston, S.C., USA) ranges between 4400 and 6500 [31,32], although there is little variation in the other analytical results [31]. The Indulin AT used in the present work was purified by dissolving in aqueous alkaline solution, from which it was recovered by acidification as a precipitate that was thoroughly washed with distilled water. After air-drying, the kraft-lignin powder was obtained in ~67% yield. 

Maple GVL lignin was provided by A.H. Motagamwala and G.W. Huber, Chemical & Biological Engineering, University of Wisconsin. Gamma-valerolactone (GVL) dissolution of lignocellulosic feedstocks from diverse sources enables fractionation of corn stover, hardwood, and softwood into relatively pure cellulose along with sugars from hemicelluloses, and lignin in which structures close to the native biopolymer are preserved [30]. A homogeneous liquid mixture of mildly acidic ~80:20 GVL:water can thermocatalytically saccharify maple wood lignocellulose as the biomass is subjected to complete dissolution of carbohydrates and lignin within 30 min at 120 °C [33]. A soluble lignin stream from maple yielded a co-product GVL lignin exhibiting a broad molecular weight distribution (*M_w_* 8000 with polydispersity index 10.7), which was suitable (after precipitation) for producing polymeric materials approaching 100 wt% in lignin content.

1,4-Anthraquinone, 9,10-anthraquinone (Alfa Aesar, Tewksbury, MA, USA), 3,5-dinitroaniline, 1,8-dinitroanthraquinone, *m*-dinitrobenzene, 4-nitroaniline, polycaprolactone, tetrabromobisphenol-A (TBBP-A) (Sigma-Aldrich, St. Louis, MO, USA), polyacrylamide, poly(ethylene oxide-*b*-1,2-butadiene-*b*-ethylene oxide) (EBE) (Polymer Source, Montreal, QC, Canada), and poly(trimethylene glutarate) (Scientific Polymer Products, Ontario, NY, USA) were used as received without further purification.

### 3.2. Casting

A 0.8 g quantity of kraft lignin or GVL lignin, with or without another blend component, was dissolved in 4.0 mL dimethyl sulfoxide (DMSO) to produce a homogeneous solution. Kraft-lignin-containing solutions were filtered through a 4–5.5 μm pore-size fritted disc. All solutions were degassed at 70 °C in 10 × 20 mm Teflon molds under reduced pressure in a vacuum oven, whereafter the temperature was raised stepwise at atmospheric pressure from 120 °C to 150 °C or 180 °C using a protocol that resulted in >99% DMSO removal. In the process, the temperature approached and/or exceeded the glass transition temperature of the lignin preparation or lignin-based blend. The resulting rectangular test piece was filed to a 1-mm thick dog-bone-shaped test specimen, of which the typical distance between shoulders was 6–7 mm and the width was 5 mm.

Alternatively, 0.6 g BML, with or without another blend component, was dissolved in 4.0 mL DMSO to produce a homogeneous solution that was transferred to a 10 × 20 mm Teflon mold for degassing under reduced pressure in a vacuum oven at 50 °C for 15 min. The BML test pieces were formed by solution-casting at 150 °C for 24 h and then 180 °C for 3 h, whereafter they were filed to the same shape and dimensions as the kraft and GVL lignin-based materials. 

### 3.3. Tensile Testing 

The tensile behavior of the lignin-based polymeric material or plastic (in the form of a dog-bone-shaped test piece) was characterized by means of a stress–strain curve measured at 50% relative humidity with an Instron model 5542 unit fitted with a 500 N static load cell. Serrated jaws were used to hold all test pieces in place. No tensile test was initiated until the load reading had become stable. A crosshead speed of 0.05 mm min^−1^ was employed with specimen gauge lengths of 6–7 mm. Young’s modulus (E) and the stress (σ_max_) and strain (ε_σ,max_) at fracture were calculated on the basis of initial sample dimensions. Test pieces with the same composition that had been produced under the same casting conditions exhibited ~2.5% variations in tensile strength. Tensile tests were performed between 6 and 20 days after casting the test pieces that had been stored under ambient conditions.

## 4. Conclusions

Material continuity is the central issue in creating functional plastics with very high (>90 wt%) lignin contents. The macromolecular species in these plastics are not individual lignin components but rather associated complexes with two distinct domains. The integrity of the inner domain is largely upheld by noncovalent interactions between cofacially-offset aromatic rings, while the peripheral domain possesses a greater number of edge-on aromatic ring arrangements. During casting, material continuity develops through interdigitation between peripheral domains of adjoining complexes.

When, through chemical changes, the peripheral domains of associated complexes in particular lignin preparations become less capable of interdigitation, easing can be achieved with suitable plasticizers. Alternatively, blending with small quantities of other lignin derivatives may enhance peripheral-domain malleability to an extent that enables continuity to be established between more rigid lignin complexes. Properly matched lignin–lignin blends can produce lignin-only plastics that go well beyond polystyrene in tensile strength.

Formulations that, through solution-casting, produce functional polymeric materials with very high lignin contents can act as starting points for optimizing the injection-molding of plastic parts and components.

## Figures and Tables

**Figure 1 molecules-24-04611-f001:**
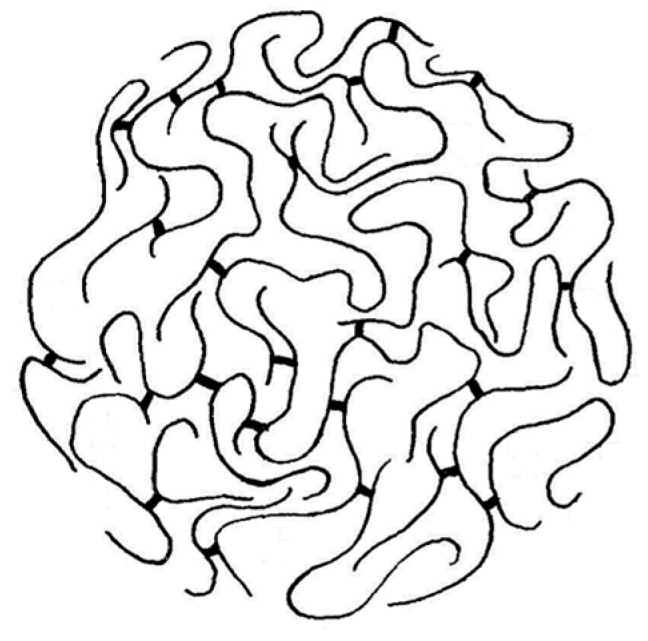
Adaptation of the macromolecular “microgel” structure proposed in 1960 [8] to describe a ligninsulfonate crosslinked polymer chain (sulfonate groups omitted).

**Figure 2 molecules-24-04611-f002:**
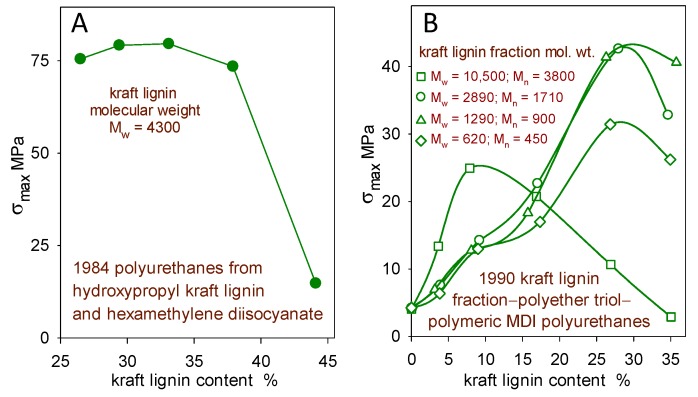
Effect of softwood kraft lignin content on the tensile strength (σ_max_) of (**A**) hydroxypropyl kraft lignin–hexamethylene diisocyanate polyurethanes [10] and (**B**) kraft lignin–polyether triol–polymeric MDI polyurethanes (0.9 overall NCO/OH reactant ratio) from four kraft lignin fractions [11].

**Figure 3 molecules-24-04611-f003:**
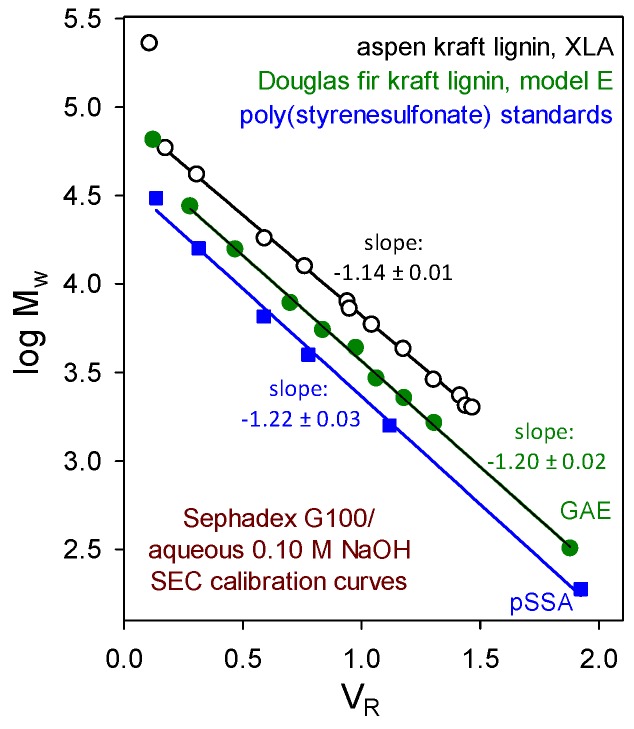
Sephadex G100/aqueous 0.10 M NaOH size-exclusion chromatographic absolute molecular-weight calibration curves determined by sedimentation equilibrium studies of paucidisperse kraft lignin fractions: aspen kraft lignin analyzed by Beckman XLA ultracentrifuge [13] and Douglas fir kraft lignin analyzed by Beckman model E instrument [14] compared with the elution behavior of poly(styrenesulfonate) standards.

**Figure 4 molecules-24-04611-f004:**
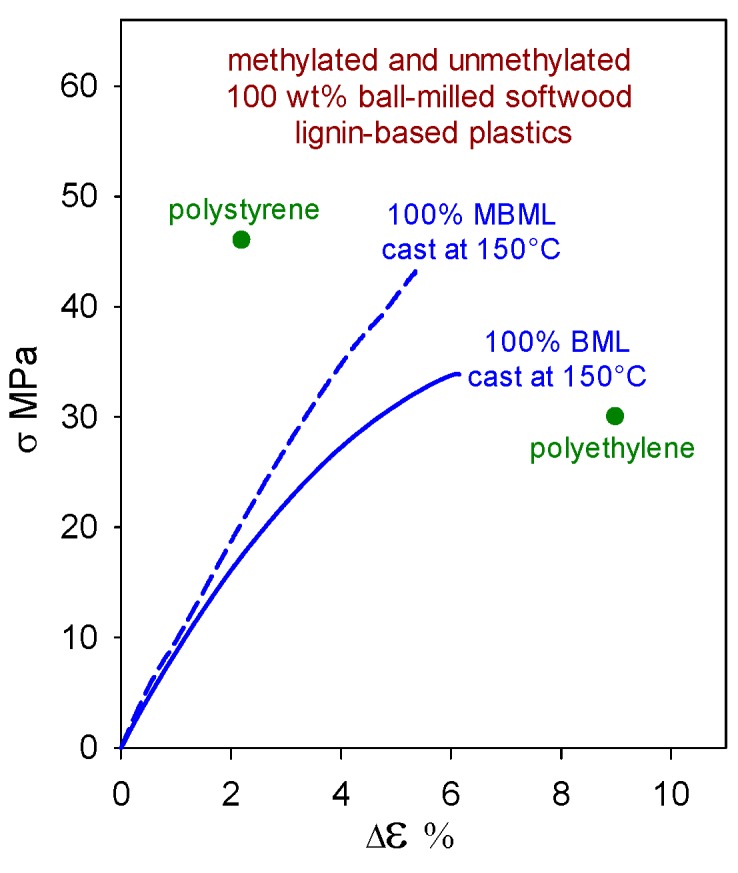
Tensile behavior of polymeric materials composed solely of unmethylated (BML) and methylated (MBML) softwood ball-milled lignin [3].

**Figure 5 molecules-24-04611-f005:**
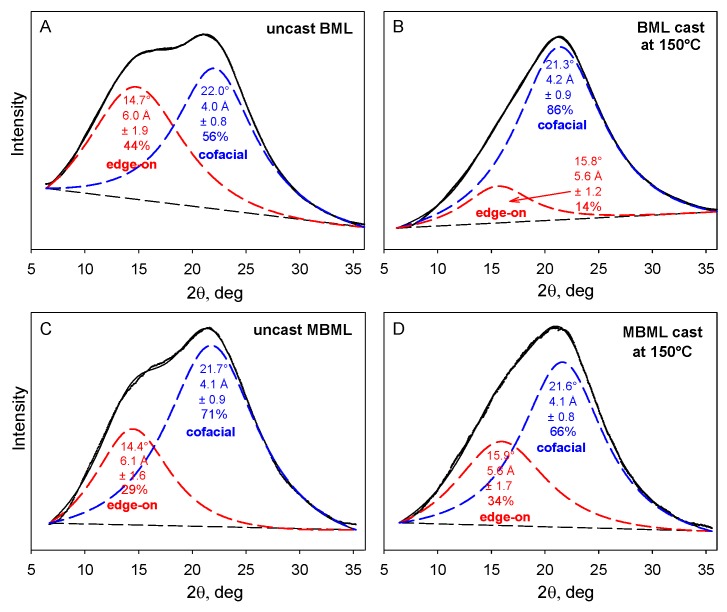
X-ray powder diffraction patterns of uncast and cast polymeric materials based on unmethylated and methylated ball-milled softwood lignins. (**A**) uncast and (**B**) cast unmethylated ball-milled lignins (BMLs); (**C**) uncast and (**D**) cast ball-milled lignin successively methylated with dimethyl sulfate and diazomethane (MBML) [3].

**Figure 6 molecules-24-04611-f006:**
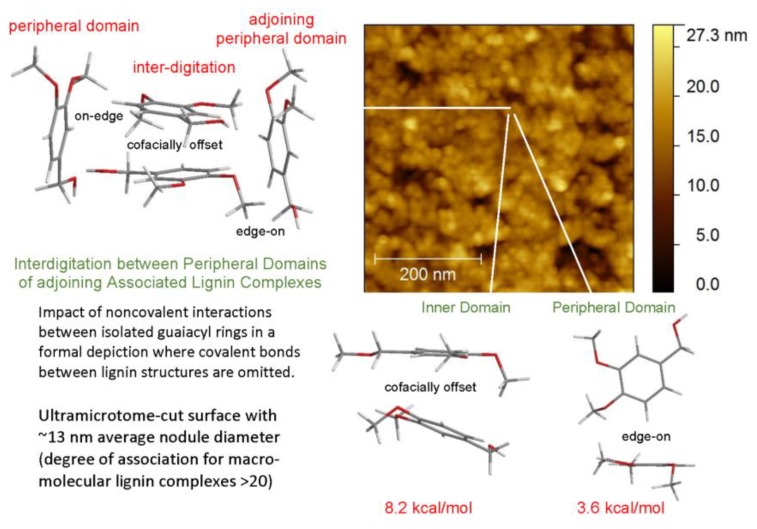
AFM height image of associated lignin complexes on 100 wt% methylated ball-milled lignin-based plastic surface [17].

**Figure 7 molecules-24-04611-f007:**
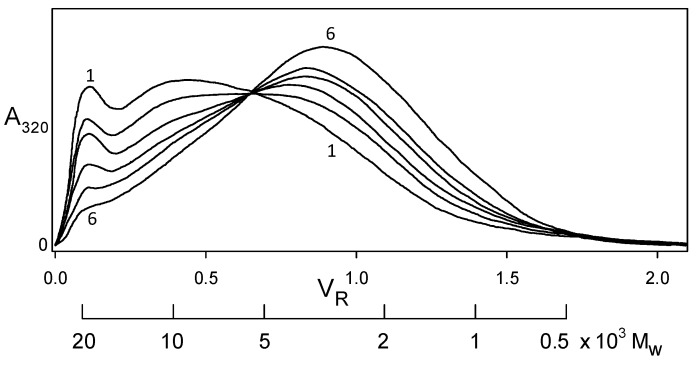
Apparent molecular weight distributions representing a homologous series of kraft lignin samples secured by desalting after association and dissociation in aqueous alkaline solutions [21] for (**1**) 300 h, (**2**) 144 h, and (**3**) 48 h at 170 gL^−1^ in 1.0 M ionic strength aqueous 0.40 M NaOH; (**4**) 0 h, (**5**) 144 h, and (**6**) 644 h at 0.50 gL^−1^ in aqueous 0.10 M NaOH. Sephadex G100/aqueous 0.10 M NaOH elution profiles monitored at 320 nm.

**Figure 8 molecules-24-04611-f008:**
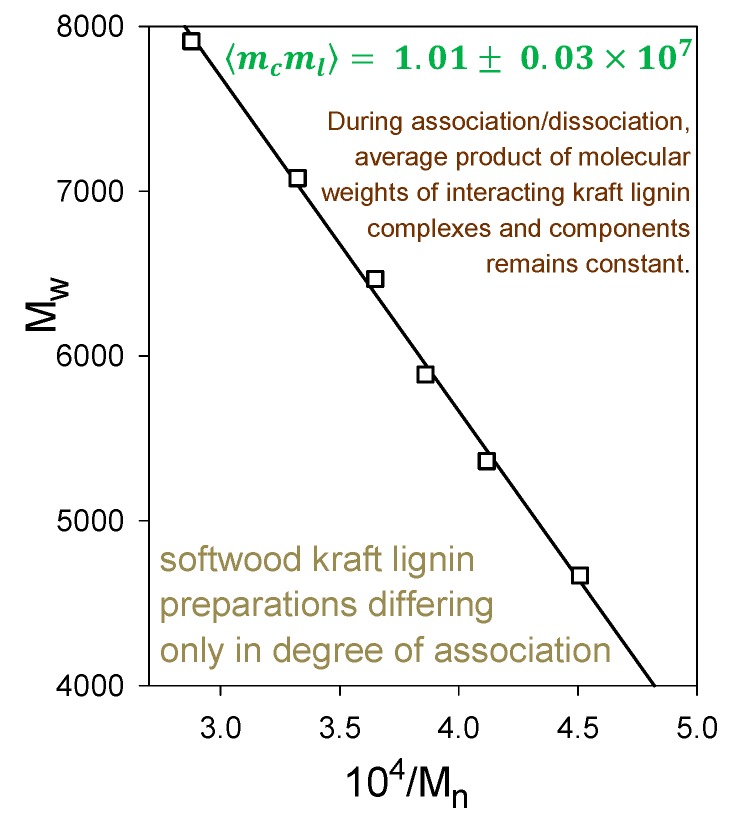
Relationship between *M_w_* and *M_n_* during associative/dissociative processes in aqueous alkaline solutions between molecular kraft lignin species leading to a series of samples with different degrees of association [21] in Figure 7, desalted through Sephadex LH20/aqueous 35% dioxane.

**Figure 9 molecules-24-04611-f009:**
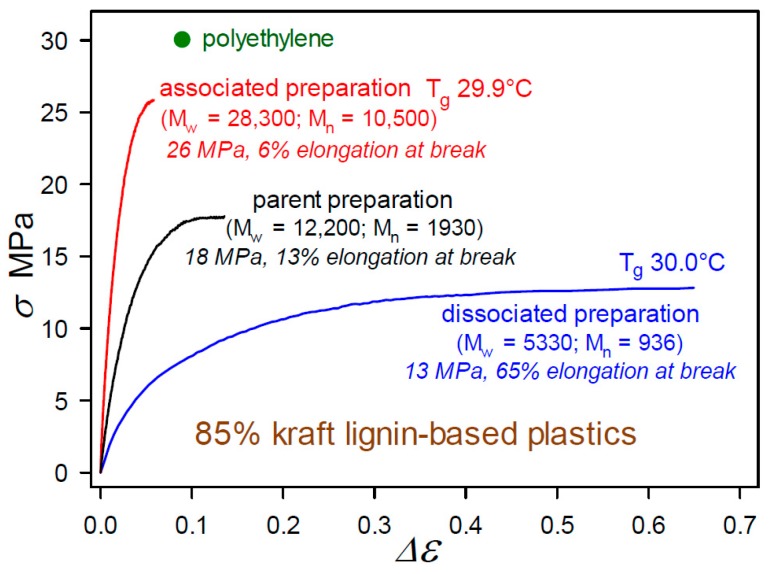
Tensile behavior to fracture of thermoplastics containing 85 wt% kraft lignin preparations that differ only in degree of intermolecular association [24] in biphasic blends with 12.6% poly(vinyl acetate), 1.6% diethyleneglycol dibenzoate, and 0.8% indene.

**Figure 10 molecules-24-04611-f010:**
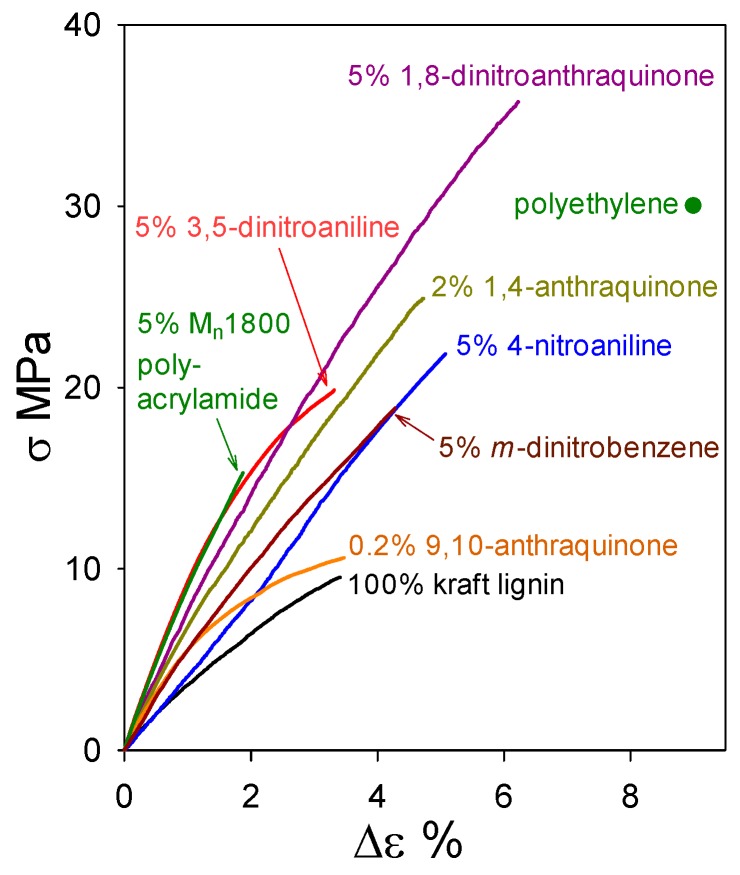
Tensile behavior of polymeric materials composed of industrial softwood kraft lignin alone and with 95–98 wt% kraft-lignin contents.

**Figure 11 molecules-24-04611-f011:**
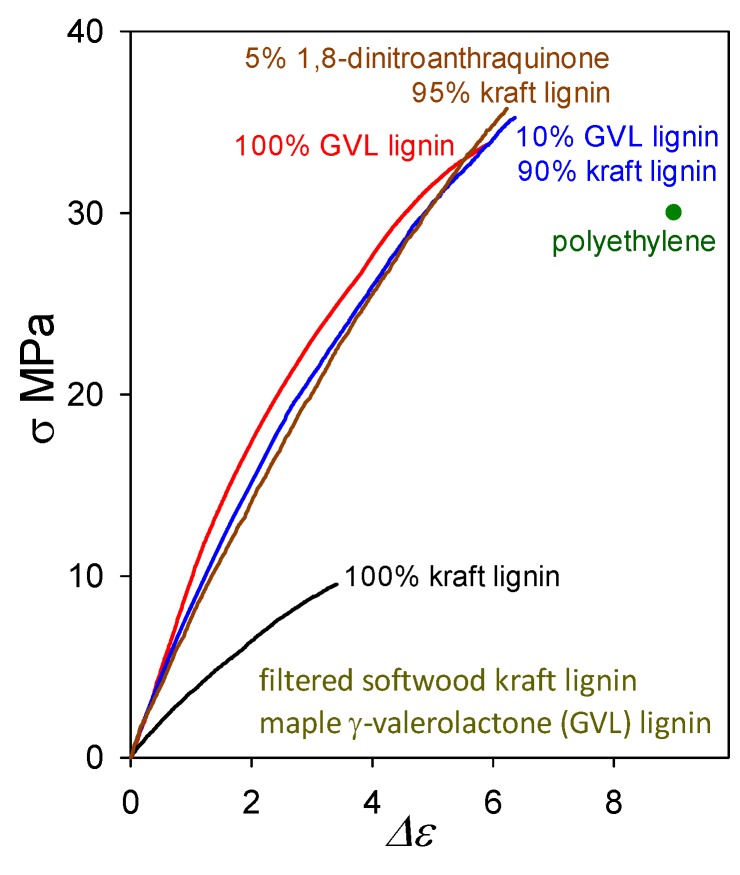
Tensile behavior of plastics based on softwood kraft lignin alone and in blends with 1,8-dinitroanthraquinone and maple GVL lignin, compared with plastics containing maple GVL lignin alone.

**Figure 12 molecules-24-04611-f012:**
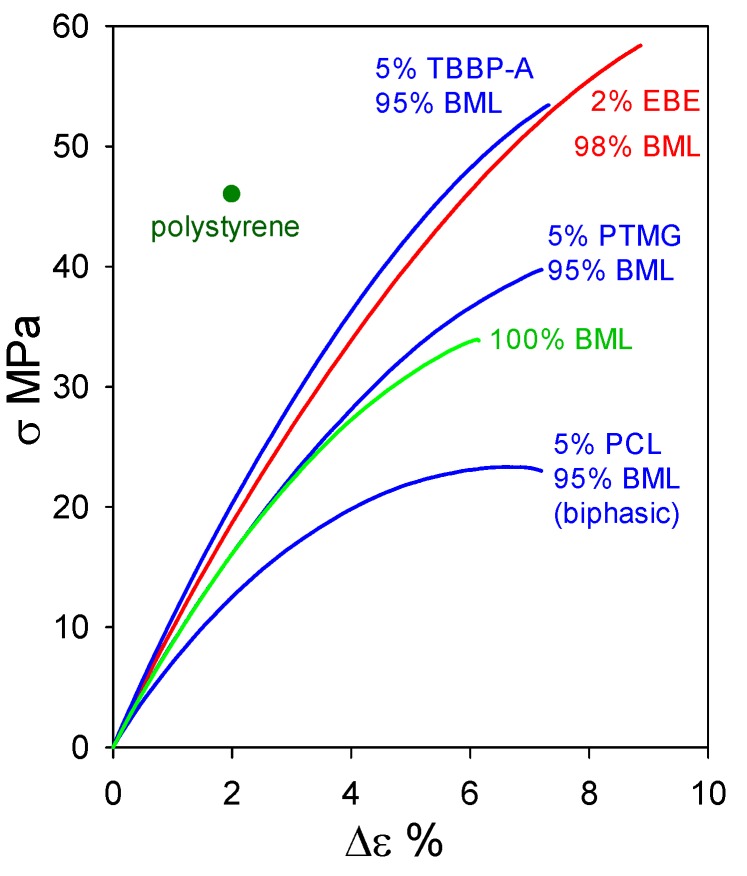
Tensile behavior of polymeric materials composed of ball-milled lignin alone and in blends with polycaprolactone (PCL), poly(ethylene oxide-*b*-1,2-butadiene-*b*-ethylene oxide) (EBE), poly(trimethylene glutarate) (PTMG), and tetrabromobisphenol A (TBBP-A).

**Figure 13 molecules-24-04611-f013:**
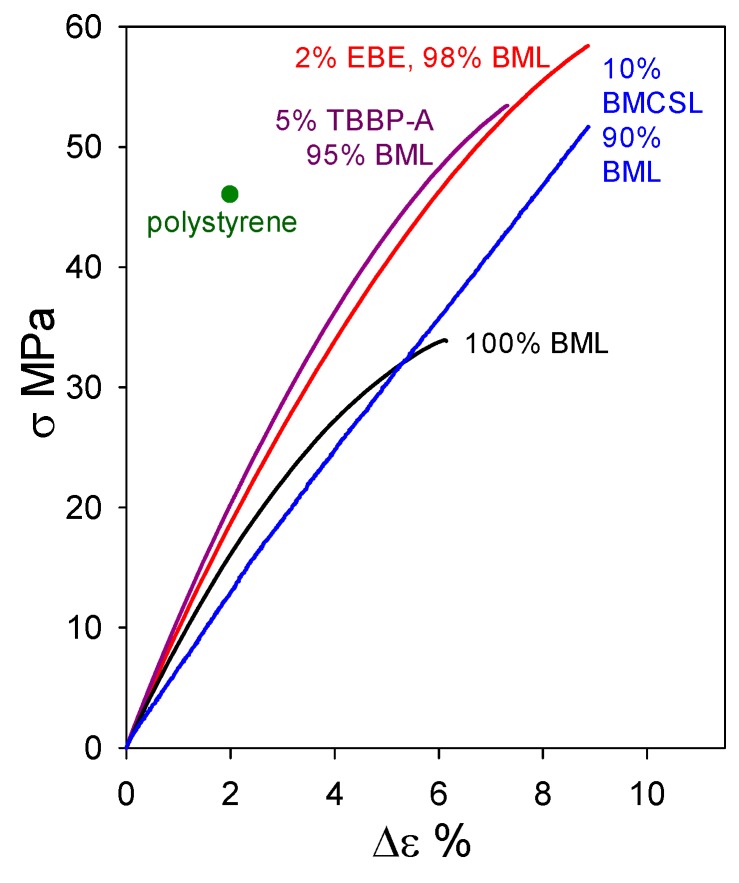
Tensile behavior of plastics based on ball-milled softwood lignin alone and in blends with ball-milled corn-stover lignin (BMCSL), poly(ethylene oxide-*b*-1,2-butadiene-*b*-ethylene oxide) (EBE), and tetrabromobisphenol A (TBBP-A).

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
