# Peer review of "Lignin-Only Polymeric Materials Based on Unmethylated Unfractionated Kraft and Ball-Milled Lignins Surpass Polyethylene and Polystyrene in Tensile Strength"

_molecules, 2019, doi:10.3390/molecules24244611_

Round 1
Reviewer 1 Report
The manuscript is a very interesting piece of work that sheds light on the preparation of lignin rich polymeric films with very interesting mechanical properties overcoming those of commodities such as polystyrene and polyethylene. The work is well structured and written and may be of interest for a wide readership, therefore deserving publications.
Only a few minor issues/curiosities are listed below:
1) the abstract is quite long and does not fully refletcs the content of the work. It is suggested to better highlight the main outcomes to catch the readers' attention.
2) Please revise the caption of Figure 2: polyurethanes10 and σmax (as subscript).
3) Please add the E and εbreak values for all the curves depicted in Figure 9.
4) The prepared "lignin-only" films show very interesting mechanical properties, but what about the possibility to use these materials as substitutes of plastic commodities (e.g. shaping into 3D objects and so on)? Do the authors Envision any possible applicability on larger scale?
Reviewer 2 Report
This is an intersting paper. I was suprised by the performance of all-lignin plastics. However, i do have a few quesions for the authors. The associative and dissociative behaviours were most studied in solutions, how were those translated to solid materials? Interpretaion of the XRD data was not convincing at present. The authors may need present more evidence. I don't understand the AFM data, how was that related to the two different configurations?
Figure 8, please elaborate on " degree of association".
Reviewer 3 Report
The authors developed lignin plastics (over 90 wt.%) with excellent mechanical performance which is even higher than that of two common polymers, polystyrene and polyethylene. By using complex association of lignin domains with non-covalent interactions and utilizing chemical changes, the formation of macromolecular structures with continuous lignin complexes resulted in rigid materials. The method developed in this report will be useful and establish fundamental background to modify lignin for high performance plastics. The manuscript was well written and is recommended to publish in Molecules as it is.
Author Response
Reviewer 3 says that "the manuscript is well written and is recommended to publish in Molecules as it is". No questions to the authors were asked.